# Air quality observations onboard commercial and targeted Zeppelin flights in Germany - a platform for high-resolution trace-gas and aerosol measurements within the planetary boundary layer

Ralf Tillmann[1], Georgios I. Gkatzelis[1], Franz Rohrer[1], Benjamin Winter[1], Christian Wesolek[1], Tobias Schuldt[1], Anne C. Lange[1], Philipp Franke[1], Elmar Friese[1], Michael Decker[1], Robert Wegener[1], Morten Hundt[2], Oleg Aseev[2], Astrid Kiendler-Scharr[1]

[1] Institute of Energy and Climate Research, IEK-8: Troposphere, Forschungszentrum Jülich GmbH, Jülich, Germany

[2] MIRO Analytical AG, Wallisellen, 8304, Switzerland

*Correspondence to*: Ralf Tillmann (r.tillmann@fz-juelich.de)

**Abstract.**

A Zeppelin airship was used as a platform for in-situ measurements of greenhouse gases and short-lived air pollutants within the planetary boundary layer (PBL) in Germany. A novel quantum cascade laser-based multi-compound gas analyzer (MIRO Analytical AG) was deployed to simultaneously measure in-situ concentrations of greenhouse gases ($CO_2$, $N_2O$, $H_2O$, and $CH_4$) and air pollutants (CO, NO, $NO_2$, $O_3$, $SO_2$, and $NH_3$) with high precision at a measurement rate of 1 Hz. These measurements were complemented by electrochemical sensors for NO, $NO_2$, $O_x$ ($NO_2+O_3$), and CO, an optical particle counter, temperature, humidity, altitude, and position monitoring. Instruments were operated remotely without the need for on-site interactions. Three two-week campaigns were conducted in 2020 comprising commercial passenger as well as targeted flights over multiple German cities including Cologne, Mönchengladbach, Düsseldorf, Aachen, Frankfurt, but also over industrial areas and highways.

Vertical profiles of trace gases were obtained during the airship landing and take-off. Diurnal variability of the Zeppelin vertical profiles was compared to measurements from ground-based monitoring stations with a focus on nitrogen oxides and ozone. We find that their variability can be explained by the increasing nocturnal boundary layer height from early morning towards midday, an increase in emissions during rush hour traffic, and the rapid photochemical activity midday. Higher altitude (250–450 m) $NO_x$ to CO ratios are further compared to the 2015 EDGAR emission inventory to find that pollutant concentrations are influenced by transportation and residential emissions as well as manufacturing industries and construction activity. Finally, we report $NO_x$ and CO concentrations from one plume transect originating from a coal power plant and compare it to the EURAD-IM model to find agreement within 15%. However, due to the increased contribution of solar and wind energy and/or the impact of lockdown measures the power plant was operated at max. 50% capacity; therefore, possible overestimation of emissions by the model cannot be excluded.

## 1 Introduction

Favorable meteorological conditions can trigger severe pollution episodes in which anthropogenic emissions of pollutant concentrations accumulate and drastically exceed the World Health Organization (WHO) guideline values. Meteorologically induced air pollution is consistently observed globally in Asia (He et al., 2017; Li et al., 2019; Cai et al., 2017; Zhao et al., 2019), America (Jury, 2020; Zhao et al., 2011; Lin and McElroy, 2010), and Europe (Dupont et al., 2016; Pernigotti et al., 2012) even during periods when certain anthropogenic emission sectors are diminished (Gkatzelis et al., 2021a). With air quality being the number one environmental health risk globally (WHO, 2021; Lelieveld et al., 2015), there is an increasing need to monitor pollutant concentrations in time and amplitude in order to identify the driving factors for degraded air quality. An essential first step towards this goal is to accurately determine the effect of local meteorological parameters such as surface relative humidity, wind speed, turbulence, and planetary boundary layer (PBL) depth development on pollutant concentrations. Up to date, various studies highlight the need for accurate PBL depth data as they pose the most uncertain parameter for efficient air quality forecasts (e.g., Dupont et al., 2016; Lin and McElroy, 2010; Silcox et al., 2012; Horel et al., 2016). Vertical mixing of air tracers within the PBL can influence their tropospheric distributions with a turbulent mixed layer leading to a more uniform vertical distribution and a stable boundary layer resulting in greater vertical gradients.

Numerous European ground-based networks e.g., the European Environment Agency, EEA, together with the European Monitoring and Evaluation Programme, EMEP (http://ebas.nilu.no), or from infrastructures such as the Aerosols, Clouds and Trace gases Research Infrastructure (ACTRIS; https://www.actris.eu/) provide data for criteria pollutant concentrations worldwide; however, there is still a lack of information along the vertical. Satellite retrievals allow global coverage of pollutant concentrations but only obtain the nadir total column with limited information on the vertical distribution of pollutant concentrations (e.g., Veefkind et al., 2012). On the other hand, aircraft campaigns provide pollutant concentrations at various altitudes (e.g., Molina et al., 2010; Ryerson et al., 2013; Benedict et al., 2019); however, obtaining vertical profiles is challenging and the data availability is limited due to high rental aircraft costs. A way to overcome such a limitation has been to deploy instrumentation in commercial airliners as has been done in the last decades by the In-service Aircraft for a Global Observing System (IAGOS; https://www.iagos.org; Marenco et al., 1998; Petzold et al., 2015). Such measurements provide regular data on the PBL dynamics but are limited to areas in proximity to airport locations during the aircraft's landing and take-off (Boschetti et al., 2015). Commercial airborne measurements have also been extended to routine helicopter flights to monitor vertical profiles for pollutant concentrations in Utah, USA (Crosman et al., 2017). Balloon-borne (e.g., Ouchi et al., 2019) and small unmanned aerial vehicles i.e., drones (Villa et al., 2016) are also frequently used for vertical profile measurements; however, they cover a limited number of pollutants due to weight restrictions. Finally, ground-based LIDAR measurements can provide a diagnosis on the PBL height and vertical concentration profiles but are often limited to only one pollutant (e.g., Dang et al., 2019).

A Zeppelin is an ideal airborne platform to capture the vertical distribution of pollutant concentrations and gain insights into their origin and emission sources (Lampilahti et al., 2021; Li et al., 2014; Nieminen et al., 2015). It offers enough room to deploy equipment and flies precisely and slowly at desired heights. Such airborne

measurements provide unique opportunities to compare to modeling efforts and evaluate and update air quality forecasts and emission inventories for single point sources.

Here, we present commercial and targeted Zeppelin flights in Germany using state-of-the-art instrumentation to investigate the vertical, spatial, and temporal distribution of pollutant concentrations including nitrogen oxides (NO, $NO_2$, and $NO_x$), ozone ($O_3$), carbon monoxide (CO), carbon dioxide ($CO_2$), and others. We compare these results to observations from ground-based monitoring stations and emission inventory estimates. Finally, we report emissions from a coal power plant and compare our measurements to the EURAD-IM model hindcast (Elbern et al., 2007).

## 2 Methods

### 2.1 Zeppelin platform

The airborne platform used in this study was the Zeppelin New Technology (NT) developed by Zeppelin Luftschifftechnik GmbH & CO. KG (ZLT) in Friedrichshafen, Germany, in 1997. Zeppelin NT is an economical airship with a length of 75 m, a diameter of about 14 m, and a maximum payload of around 1.8 tons. It offers a unique combination of capabilities not available in other airborne platforms including a high scientific payload, high maneuverability in all directions due to a vectored thrust propulsion system, flight speeds from 0–115 km/h, a horizontal reach of up to more than 600 km, operating altitude of 20–1500 m, and a maximum flight endurance of 15 h.

Figure 1A shows the Zeppelin flights over Germany and Figure 1B the vertical and diurnal distribution of these flights. Detailed information on the take-off and landing times, airport locations, and flight paths are provided in Table 1. In this study, 14 days of commercial flights, 4 days of targeted flights, and 6 days of transect flights with overall 172 take-offs and landings were performed and analyzed. The Zeppelin flew over various cities, including Cologne, Mönchengladbach, Düsseldorf, Aachen, Jülich, Frankfurt, but also over industrial areas, and highways. The majority of the measurements ranged from 200 to 450 m in altitude. Measurements below 200 m were predominantly during the Zeppelin landing and take-off periods and higher altitude measurements above 400 m were during transect and targeted flights. Flights were distributed in summer 2020 to 9, 7, and 10 flight days in May, June, and September, respectively, ranging from 3–10 flight hours per day. Four airports were chosen to refuel the Zeppelin, namely, Friedrichshafen, Bonn-Hangelar, Mönchengladbach, and the Bad Homburg airfield.

Zeppelin NT has been previously used as an airborne platform fully equipped with instrumentation to conduct atmospheric research during the PEGASOS project (Li et al., 2014; Nieminen et al., 2015). Here, measurements were predominantly performed during commercial passenger flights providing low costs but limited space to deploy instrumentation. Two main instrument setups were fitted in the cabin of the Zeppelin: the MIRO instrument and the hatch box with diverse low-cost sensors as discussed in the following sections.

**Table 1:** Zeppelin flight details.

| Date | Local start time | Local end time | Airport Code | Flight Details |
|------|-----------------|----------------|--------------|----------------|
| 29.04.2020 | 8:17:00 | 15:00:00 | FDH– BNJ | Friedrichshafen - Bonn/Hangelar |
| 05.05.2020 | 9:43:00 | 16:22:00 | BNJ | Köln & Mönchenlagdbach |
| 06.05.2020 | 7:14:00 | 14:30:00 | BNJ | Weisweiler |
| 07.05.2020 | 8:30:00 | 15:30:00 | BNJ | Mönchengladbach & Duisburg |
| 08.05.2020 | 7:00:00 | 11:50:00 | BNJ | Hürtgenwald & FZJ |
| 09.05.2020 | 7:29:00 | 15:23:00 | BNJ– FDH | Hangelar-Stuttgart-Friedrichshafen |
| | | | | |
| 27.05.2020 | 7:22:00 | 12:00:00 | FDH– BNJ | Friedrichshafen-Stuttgart-Bonn |
| 29.05.2020 | 8:00:00 | 18:04:00 | BNJ | Bonn-Köln-Düsseldorf-Mönchengladbach-Bonn |
| 01.06.2020 | 7:00:00 | 16:28:00 | BNJ | Bonn-Köln-Düsseldorf-Mönchengladbach-Bonn |
| 02.06.2020 | 8:00:00 | 17:46:00 | BNJ | Bonn-Köln-Düsseldorf-Mönchengladbach-Bonn |
| 03.06.2020 | 6:30:00 | 11:15:00 | BNJ | Bonn-Köln-Düsseldorf-Mönchengladbach-Bonn |
| 11.06.2020 | 8:10:00 | 17:52:00 | BNJ | Bonn-Köln-Düsseldorf-Mönchengladbach-Bonn |
| 12.06.2020 | 7:35:00 | 18:00:00 | BNJ | Bonn-Köln-Düsseldorf-Mönchengladbach-Bonn |
| 13.06.2020 | 6:15:00 | 13:45:00 | BNJ | Bonn-Köln-Bad Honnef-Bonn |
| 15.06.2020 | 6:19:00 | 12:51:00 | BNJ– FDH | Bonn-Bad Honnef-Friedrichshafen |
| | | | | |
| 02.09.2020 | 7:23:00 | 16:39:00 | FDH | Friedrichshafen - Bodensee |
| 03.09.2020 | 7:16:00 | 12:32:00 | FDH– BNJ | Friedrichshafen-Bonn/Hangelar |
| 06.09.2020 | 8:13:00 | 17:45:00 | BNJ | Bonn - Bad Honnef - Köln - Bonn |
| 07.09.2020 | 7:45:00 | 17:00:00 | BNJ | Bonn - Bad Honnef - Köln - Bonn |
| 08.09.2020 | 13:15:00 | 16:05:00 | BNJ | Bonn - Bad Honnef - Köln - Bonn |
| 10.09.2020 | 7:48:00 | 12:37:00 | BNJ– BadH | Bonn - Frankfurt - Bad Homburg |
| 11.09.2020 | 9:10:00 | 17:00:00 | BadH | Bad Homburg - Frankfurt - Bad Homburg |
| 12.09.2020 | 7:08:00 | 16:23:00 | BadH | Bad Homburg - Frankfurt - Bad Homburg |
| 13.09.2020 | 07:07:00 | 16:22:00 | BadH | Bad Homburg - Frankfurt - Bad Homburg |
| 14.09.2020 | 06:19:00 | 11:14:30 | BadH– FDH | Ban Homburg - Friedrichshafen |

*FDH: Friedrichshafen, BNJ: Bonn-Hangelar, BadH: Bad Homburg airfield*

110

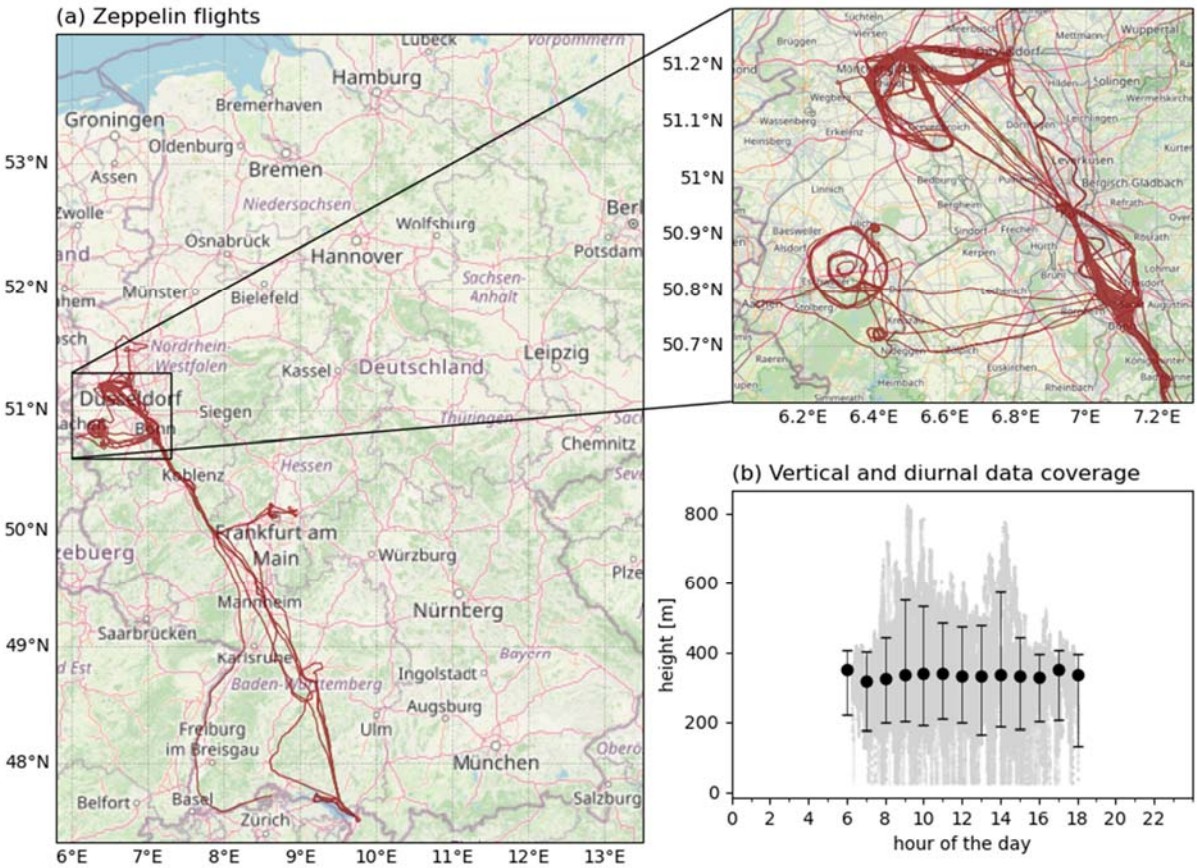

**Figure 1:** (a). The Zeppelin flight tracks over Germany (© OpenStreetMap contributors 2021. Distributed under the Open Data Commons Open Database License (ODbL) v1.0.) and (b). their vertical and diurnal coverage in UTC.

## 2.2 MIRO instrument

We deployed a MIRO MGA[10]-GP multi-compound gas analyzer, a newly available commercial instrument (MIRO Analytical AG, Wallisellen, Switzerland). The analyzer measures ten trace gases NO, $NO_2$, $O_3$, $SO_2$, CO, $CO_2$, $CH_4$, $H_2O$, $NH_3$, $N_2O$ with a time resolution of 1s and precisions (1σ) as summarized in Table 2. The stated precisions were determined by Allan-Werle-Variance (Werle et al., 1993). A detailed description of the measurement principle and the instrument's data processing and characterization of the instrument can be found elsewhere (Liu et al., 2018, Hundt et al., 2018).

**Table 2:** Instrumentation onboard the Zeppelin airship.

| Installation | Method | Parameter | Time resolution (response time) | Precision | Reference |
|---|---|---|---|---|---|
| *Cabin Rack* | Mid-infrared (MIR) direct laser absorption spectroscopy | $O_3$ | 1 s | ± 0.6 ppb (1 σ) | MIRO AG |
| | | NO | | ± 0.3 ppb (1 σ) | |
| | | $NO_2$ | | ± 0.04 ppb (1 σ) | |
| | | CO | | ± 0.04 ppb (1 σ) | |
| | | $SO_2$ | | ± 1.7 ppb (1 σ) | |
| | | $NH_3$ | | ± 0.05 ppb (1 σ) | |
| | | $H_2O$ | | ± 12 ppm (1 σ) | |
| | | $CH_4$ | | ± 0.7 ppb (1 σ) | |
| | | $N_2O$ | | ± 0.08 ppb (1 σ) | |
| | | $CO_2$ | | ± 750 ppb (1 σ) | |
| *Hatch box* | Capacitive polymer sensor chip | RH | 1s (4 s) | ± 2 % RH | |
| | | T | 1 s (5 s) | ± 0.3°C | |
| | Electrochemical sensor | NO | 1 s (< 45 s) | ± 15 ppb (2 σ) | Alphasense NO-B4 |
| | | CO | 1 s (< 30 s) | ± 4 ppb (2 σ) | Alphasense CO-B4 |
| | | $NO_2$ | 1 s (< 80 s) | ± 15 ppb (2 σ) | Alphasense NO2-B43F |
| | | $O_x$ ($NO_2 + O_3$) | 1 s (< 80 s) | ± 15 ppb (2 σ) | Alphasense OX-B431 |
| | Optical particle counter (0.3 – 10 μm) | Aerosol particles | 1.3 s | - | Alphasense OPC- N3 |
| | GPS | Latitude, Longitude, Altitude | 5 s | - | NAVIN miniHomer 2.8 |

125

The instrument is a quantum cascade laser-based (QCL) spectrometer containing five distributed feedback (DFB) QCLs. The gas mixing ratios are measured by direct laser absorption spectroscopy of selected vibrational absorption lines of the target molecules. Beer-Lambert's law provides the relation between the light transmission and the mixing ratios of an absorbing species. To obtain the transmission spectra, the laser light is steered through an astigmatic Herriott cell to a Mercury Cadmium Telluride (MCT) detector. The Herriott cell is constantly flushed with the sampled gas providing an online in-situ measurement. The pressure in the cell was maintained at 95 hPa using a pressure controller in combination with a membrane pump. For temperature stabilization of the QCLs, an external water chiller is connected to the instrument to minimize drifts caused by ambient temperature variations. During the Zeppelin flights, the instrument's inlet was switched to a zero-air supply of $NO_x$- and $O_3$-free air by sampling cabin air through zero air cartridges for 2 min every 20 min. The obtained zero points were used to apply a background correction to the NO, $NO_2$, and $O_3$ data to reduce their drift. For consecutive zero air measurements median background drift values were 0.56 ppb, 0.34 ppb and 2.9 ppb for NO, $NO_2$ and $O_3$, respectively.

The instrument's software runs on an integrated portable computer, which offers remote access and full remote control over the analyzer and its settings if an internet connection is provided. For operation on the Zeppelin, the analyzer and its peripheral devices were integrated into a 19-inch rack as shown in Figure 2. The instrument is four rack units (18 cm) high and 61 cm deep. The sample inlet line connected to the MIRO consisted of an unheated 8 m long PFA (perfluoroalkoxy alkane) tube with an internal diameter of 4 mm. The sample air was drawn from the inlet located at the hatch box below the Zeppelin cabin at a flow rate of 1.2 lpm resulting in a residence time of around 5 s.

The performance of the MIRO with its sampling line to measure sticky molecules including $NH_3$ and $H_2O$ was further examined by laboratory measurements which mimicked the conditions during the Zeppelin flights. Fast changes of pollutant concentrations were applied to determine the response times ($t_{90}$) of the measurement system, which were 240 s and 9 s for $NH_3$ and $H_2O$, respectively (see Figure S1). This highlights the future need for a heated sampling line in order to provide quality assured data, especially for $NH_3$. We therefore omitted $NH_3$ from our discussion. For $H_2O$ and less sticky molecules response times below 9 s result in a spatial horizontal resolution of < 150 m considering a horizontal Zeppelin flight speed of 60 km/h and a vertical resolution of < 15 m for a vertical speed of 1.7 m/s. This provides the upper limits of the spatial resolution of pollutant concentration but is sufficient for the analysis included in this work. Finally, the instrument detection limit for $SO_2$ is 1.7 ppb i.e., 4.9 µg/m³ (1 σ) and the expected $SO_2$ concentrations in European urban areas are mostly below 5 µg/m³ (Henschel et al., 2013). Therefore, $SO_2$ measurements are omitted from further discussions within this paper.

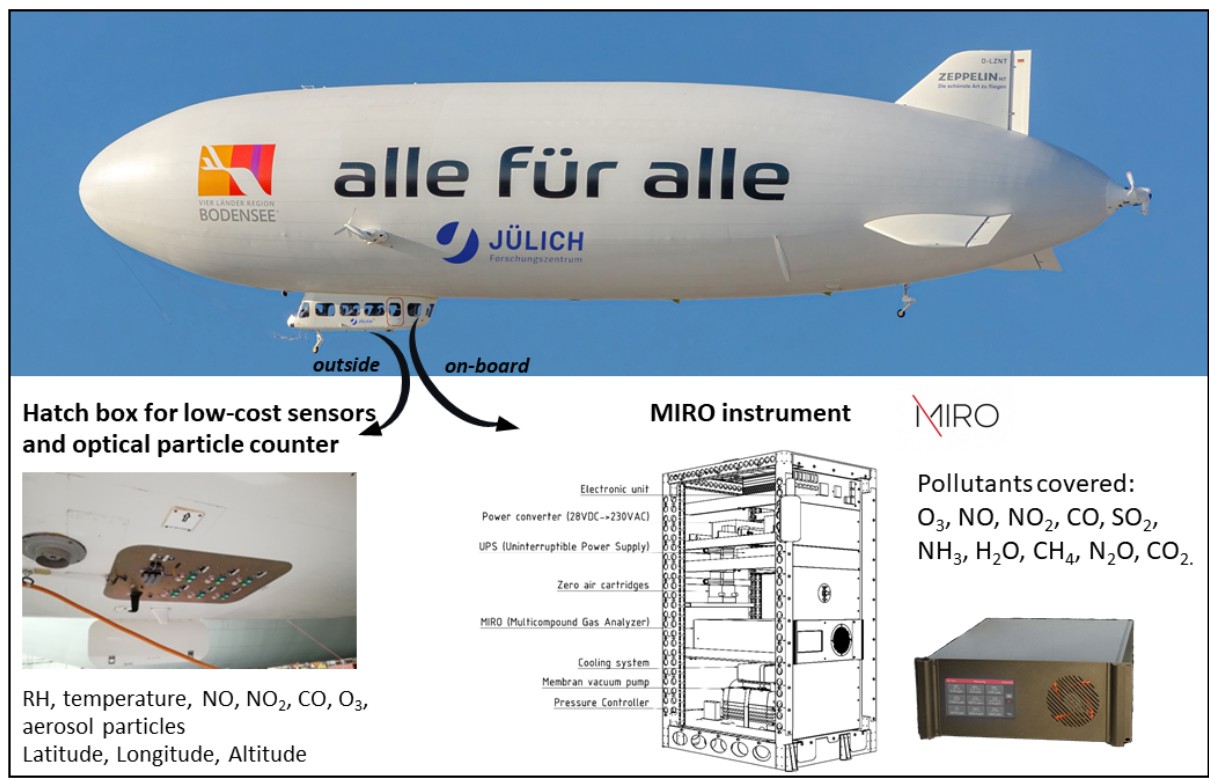

**Figure 2:** Instrumentation onboard the Zeppelin aircraft. **(**Zeppelin picture by Michael Häfner**)**

### 2.3 Hatch box for low-cost sensors and optical particle counter

Figure 2 and Figure S2 show the hatch box arrangement for multiple sensors deployed below the Zeppelin cabin. Six setups were installed each including an amperometric electrochemical gas sensor (ECS) for CO, NO, $NO_2$, and $O_X$ ($O_3 + NO_2$) measurements (Baron and Saffell, 2017), a ChipCap2 sensor for temperature (T) and relative humidity (RH) measurements, and a GPS locator for latitude, longitude and altitude measurements. Particle-phase size distribution measurements were performed by two optical particle counters (OPCs). Currently, the optical counter has been used based on the available instrument recommendations with and without an isokinetic inlet. The isokinetic inlet has been a 3-D-printed L-shaped inlet of 10 cm length and an internal diameter of 5 mm. The second OPC was used without any inlet line, sampling perpendicular to the flight direction. No further sample preparation or quality assurance has been performed. We therefore exclude the OPC data in its current state from further discussion. Two Long Term Evolution (LTE) antennas were used for real-time communication to the instruments from the ground and wireless communication for onboard decisions. Details on the performance of the sensors including their time resolution, the limit of detection, and references are found in Table 2. Furthermore, the potential of the ECSs to measure nitrogen oxides is shown in Figure S3. On average, ECS $NO_x$ data are higher by 20% compared to the MIRO for concentrations above 15 ppbv which is the limit of detection for $NO_x$ measured by ECS. This makes the ECSs ideal for the identification of high $NO_x$ emission sources during the Zeppelin flights but limited in determining $NO_x$ variability at low-$NO_x$ environments. Calibrations, sensitivity analysis, and associated uncertainties of the ECSs measurements will be further discussed in a separate publication and are not

the focus of this work. In the following, all measured pollutant concentrations are acquired from the MIRO instrument.

**2.4 The EURAD-IM model**

The EURAD-IM model output was compared to the Zeppelin observations by focusing on the emissions and evolution of an industrial plume as discussed in section 3.4. Details of the model are provided by Elbern et al. (2007). Briefly, the regional emission inventory provided by the Copernicus Atmosphere Monitoring Service (CAMS) (Kuenen et al., 2014) was used and further refined using land use information. The Weather Research and Forecasting (WRF) model version 4.0.3 (Skamarock et al., 2008, Powers et al., 2017) was initialized using the global analysis of the European Centre for Medium-Range Weather Forecasts (ECMWF) to account for the meteorological effects whereas the RACM-MIM chemical mechanism (Pöschl et al., 2000) was applied to account for the effects of atmospheric chemistry on pollutant concentrations. Here, we focus on 1-h time resolution model concentrations for $NO_x$ and CO on a 1 km horizontal grid. The observational average flight height during the comparison periods was around 300 m and the model level centered at about 268 m was chosen as the closest vertical grid point in proximity to these measurements. A challenge in accurately determining the location of different emission sources in the model is that the horizontal resolution of the inventory emissions is coarser (approx. 7 km in Western Germany) than the model resolution (1 km). When refining the emissions to match the model grid, it is often hard to match single emission patterns with an operational forecast model. This was the case for the modeled industrial emission source investigated in section 3.4 that was offset by 3 km to the southeast compared to the original location of the power plant. Here, $NO_x$ concentration fields that are associated with the significant point source were reallocated to match the location of the industry and improve the comparison to observations. Reallocation of the CO concentrations was not applied since this studied power production facility had negligible CO emissions. CO background concentrations were variable due to the numerous other point sources and the longer CO lifetime.

**3 Results and Discussion**

**3.1 Vertical profiles in Frankfurt and Bonn**

Zeppelins can climb and descend slowly at confined locations to obtain the vertical distribution of pollutants. Each day, multiple flights were performed with vertical measurements obtained during the Zeppelin landing and take-off at the airports. Figure 3 shows the vertical profiles of $NO_x$ and $O_3$ at different times of the day for measurements performed nearby Frankfurt in September and Bonn in May, June, and September. For Bonn, larger variability in pollutant concentrations due to the broader seasonal coverage was not evident with measurements in May and June showing on average similar vertical trends as in September (Figure S4). Zeppelin data below 25 m were excluded from the analysis as they were affected by the Zeppelin engine exhaust emissions. In the early hours from 06:00 to 08:00 UTC, $NO_x$ mixing ratios close to the ground were higher with a median ($25^{th}$–$75^{th}$ percentile) of 8 (5–19) ppbv and 5.7 (3–20) ppbv for Frankfurt and Bonn, respectively. In Frankfurt, the median $NO_x$ concentration sharply decreased down to 1 (0.27–6) ppbv when above 125 m, whereas in Bonn, a moderate decrease to 5 (2.7–10) ppbv was observed. $O_3$ showed the opposite trend with low mixing ratios close to the ground and an $O_3$ increase above 125 m in height. During the period from 08:00 to 10:00 UTC, an increase of

NO$_x$ at all heights compared to 06:00 UTC was evident in Frankfurt, whereas in Bonn, NO$_x$ was similar to earlier hours with a slight increase of ground-level NO$_x$ to 9.2 (3.1–14.5) ppbv. In parallel, O$_3$ decreased at higher altitudes and increased closer to the ground for Frankfurt and Bonn compared to earlier hours. From 10:00 to 18:00 UTC, the lower PBL was well mixed with NO$_x$ and O$_3$ concentrations agreeing within their variability at all heights from 25 m up to 375 m. NO$_x$ concentrations decreased throughout the day down to less than 1 ppbv, whereas O$_3$ increased up to more than 60 ppbv for both Frankfurt and Bonn. Same trends were observed for other criteria pollutants including CO, CO$_2$, and CH$_4$ (Figure S5), whereas N$_2$O, H$_2$O, and NH$_3$ were less variable (Figure S6).

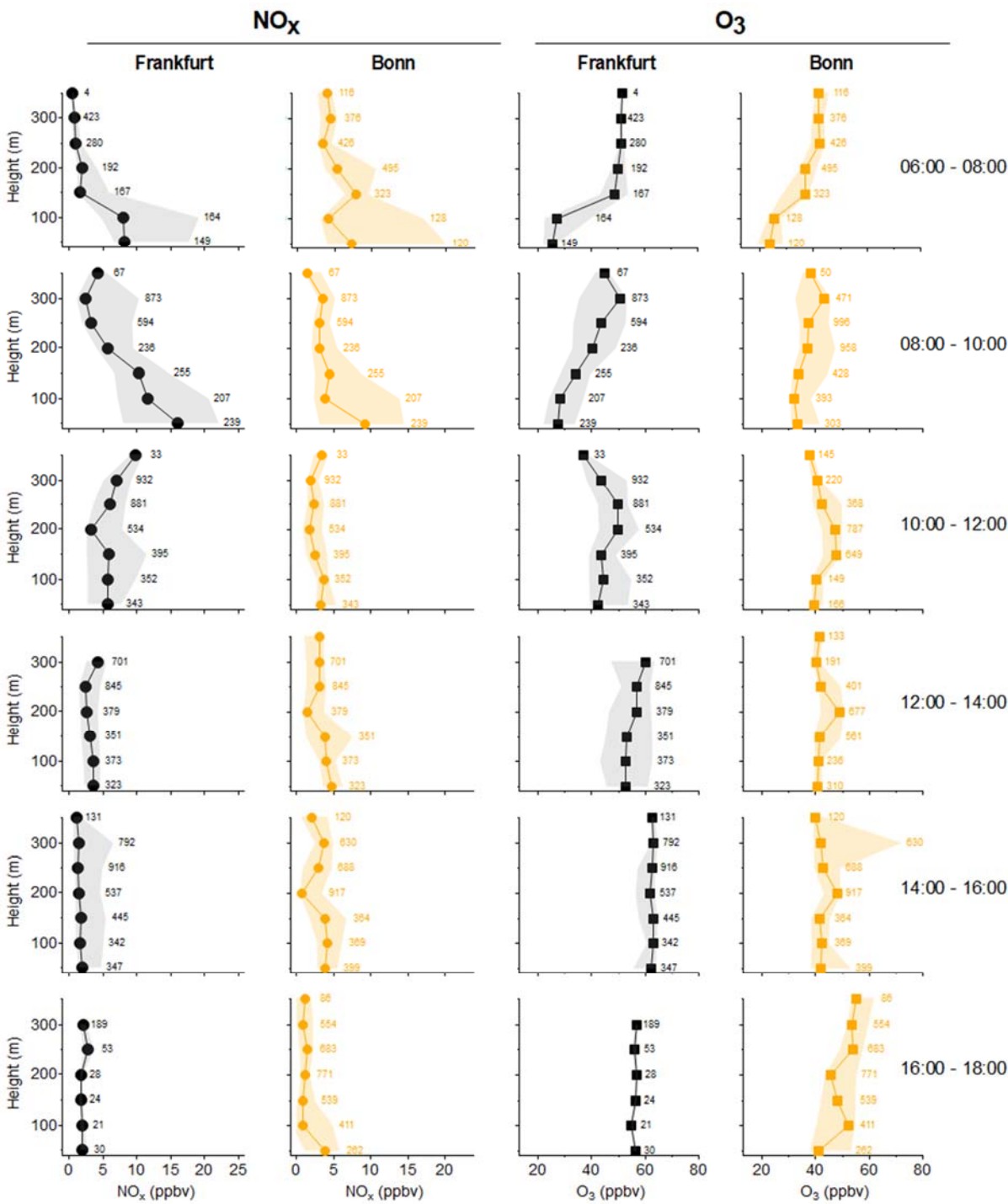

**Figure 3:** Vertical profiles for different time-periods (UTC) for $NO_x$ and $O_3$ in Bonn and Frankfurt. Circle and square markers correspond to the median $NO_x$ and $O_3$ mixing ratios, respectively. The shaded areas represent the $25^{th}$ and $75^{th}$ percentiles. Numbers correspond to the data points used to generate the $NO_x$ and $O_3$ medians.

Changes in PBL dynamics, anthropogenic emissions, and atmospheric chemistry are the drivers of the observed diurnal and vertical variability. During nighttime, a shear develops between the residual layer and the more stagnant nocturnal boundary layer that grows higher and reaches a morning maximum (see Figure S7). Shortly after sunrise, the surface heats up and a mixed layer evolves increasing with height until the former nocturnal boundary layer and the residual layer are finally fully mixed. From 08:00 to 10:00 UTC, the Zeppelin captured

the evolving convective mixed layer that developed after sunrise and reached on average up to 125 m. The increased ground-level $NO_x$ concentrations during the early hours are the result of fresh emissions from ground sources into a shallow mixed layer. As the convective mixed layer increases, ground-level $NO_x$ concentrations are expected to decrease due to dilution effects. Higher altitude concentrations increase as more concentrated ground-level $NO_x$ is distributed vertically. The morning anthropogenic emissions including rush-hour traffic increase the

ground-level $NO_x$, but dilution mitigates the level of $NO_x$ concentrations at low altitudes in Frankfurt and Bonn during the morning hours. $O_3$ is a secondary product from the interplay of $NO_x$, volatile organic compound emissions, and meteorology. Under dark conditions, $O_3$ is expected to react away and its concentration decreases, whereas during the day, it is expected to reach maximum concentrations midday when photochemistry peaks. These trends are verified by the Zeppelin flights in Frankfurt and Bonn where early morning $O_3$ titration is

followed by photochemical $O_3$ production/increase midday with uniform vertical distributions obtained after 10:00 UTC.

**3.2 $NO_x$ vertical profiles compared to ground-based monitoring stations**

The above field-derived vertical profiles show the influence of PBL dynamics in diluting pollutant concentrations but also highlight the influence of anthropogenic emissions. Comparison of these profiles to ground-based

measurements provides insights into their origin and location. Figure 4 compares the $NO_x$ vertical profiles in Frankfurt to ground-based observations from various monitoring stations (provided by Unweltbundesamt, Table S1). Twelve ground-based monitoring stations were chosen located in the broader Frankfurt metropolitan area as shown in Figure 4a. Monitoring stations in the inner city namely, Frankfurt-Höchst, Frankfurt-Niedwald, Frankfurt-Ost, Frankfurt-Schwanheim, Frankfurt-Friedberger Landstraße, and Offenbach-Untere Grenzstraße are

categorized as urban. Monitoring stations in the outer Frankfurt area including Hanau, Raunheim, Wiesbaden-Süd, Wiesbaden-Ringkirche, and Wiesbaden-Schiersteiner Straße were categorized as suburban. Finally, Kleiner Feldberg was considered a remote station at a higher altitude, 700 m above the city center of Frankfurt and 300-400 m above the Zeppelin flight track. Figure 4b shows a detailed comparison of the $NO_x$ diurnal variability of the ground stations and the Zeppelin measurements. For the urban and suburban stations, a $NO_x$ peak was evident

with an average (± 1σ) concentration of 46.3 (± 4.9) ppbv and 35.6 (± 10.1) ppbv in the morning hours. Midday, the $NO_x$ concentrations decreased due to the increasing convective mixed layer height as well as reduced traffic emissions and increased again in the evening due to rush-hour traffic reaching a maximum of 56.6 (± 25) ppbv and 41.3 (± 23.7) ppbv for the urban and suburban stations, respectively. At the remote station, the $NO_x$ concentrations were at background levels and no significant anthropogenic contribution was evident. Zeppelin

data followed the same morning increase as the urban and suburban monitoring stations with maximum $NO_x$ at

21.7, 21.3, 11.4, and 5 ppbv for measurements at 50, 100, 150, and 200 m, respectively, during the time from 08:00–10:00 UTC. For higher altitude measurements at 250 and 300 m, the $NO_x$ concentrations were the highest from 10:00–12:00 UTC with concentrations of 6.3 and 8.85 ppbv, respectively.

At lower altitudes, Zeppelin flew close to the outer urban airfield while at higher altitudes it was located closer to the city center of Frankfurt as shown in Figure 4a. It is therefore expected that measurements taken at low altitudes are comparable to those from suburban monitoring stations in particular during the early morning hours when the convective mixing layer starts evolving and the dilution of fresh emissions is less pronounced. However, ground-based (sub-)urban stations are located close to roads or even in road canyons catching fresh emissions and observed concentrations of primary emissions were higher on average than the respective low altitude Zeppelin data (Figure 4b). In the morning hours, the effect of rush hour traffic emissions on both the Zeppelin and urban/suburban stations is evident with a peak in $NO_x$ concentrations at altitudes below 200 m at 08:00–10:00 UTC. For higher altitude measurements (250–300 m) the $NO_x$ concentrations peak at 10:00–12:00 UTC which highlights the effect of PBL dynamics where morning emissions are distributed vertically to generate a well-mixed layer, resulting in a dilution of the mixed air masses. This results in a later and weaker peak of the NOx concentration at high altitudes. As the convective layer reaches altitudes above the upper flight range of the Zeppelin, this observed change in dynamic is no longer captured. In the afternoon, no significant concentration differences are observed between the different heights. The evening peak captured by the monitoring stations is only partially measured by the Zeppelin. Future Zeppelin campaigns to capture the nocturnal boundary layer development in the evening hours will provide further insights into the diurnal variability of the PBL.

These results provide evidence of effective vertical mixing in particular during the afternoon but limited to the heights captured by the Zeppelin. Future studies to convey whether mixing is as efficient at higher altitudes, that has been previously achieved by aircraft studies (e.g., Flynn et al., 2014; Flynn et al., 2016; Choi et al., 2020; Li et al., 2021), will be of great interest. Furthermore, comparison of the Zeppelin flights not only to ground-based monitoring stations, but also to modeling efforts as well as satellite measurements, would be of great value as geostationary satellites come on-line in the future.

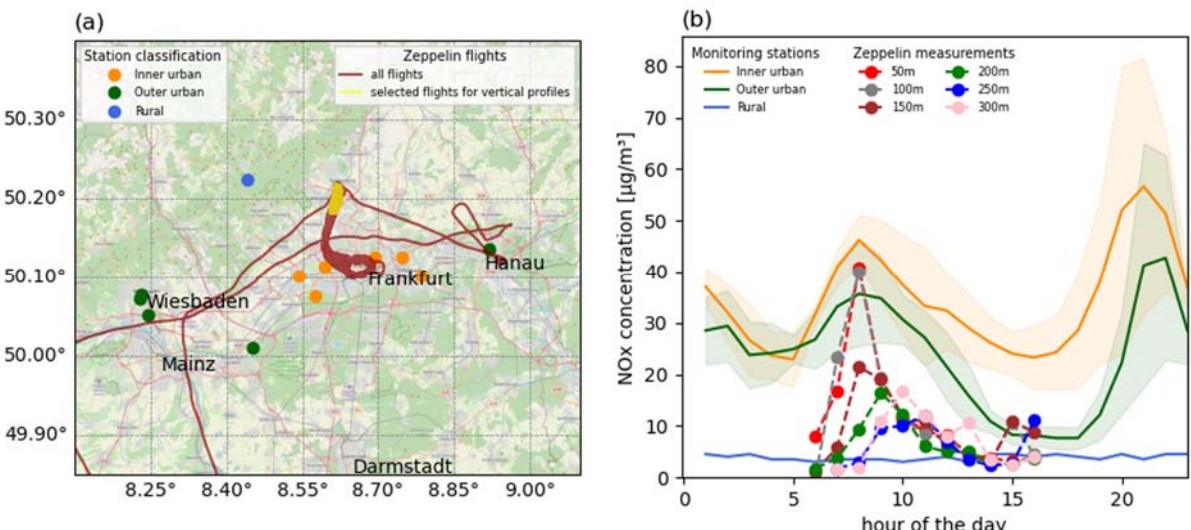

**Figure 4:** a. The map of Frankfurt overlaid with the Zeppelin flight tracks in red and yellow that are used to generate the vertical profiles shown on the right. Multiple flight paths to the city center of Frankfurt are overlayed and appear as a thick red

line. Also shown are the urban, suburban, and remote monitoring station locations with brown, green, and blue circle markers, respectively (Data provided by the Umweltbundesamt). (© OpenStreetMap contributors 2021. Distributed under the Open Data Commons Open Database License (ODbL) v1.0.) b. Average diurnal $NO_x$ concentration profiles for the urban, suburban, and remote monitoring stations and the Zeppelin diurnal profiles at different heights.

**3.3 Comparison to the EDGAR 2015 emission inventory**

Airborne Zeppelin measurements are an ideal platform to investigate pollutant concentrations on the vertical; however, the majority of flight hours were at heights ranging from 250-450 m. During these periods, the Zeppelin flew over various cities, including Cologne, Mönchengladbach, Düsseldorf, Aachen, Frankfurt, but also over industrial areas, and highways. This provided the opportunity to better characterize anthropogenic emissions and compare observations to emission inventory estimates for Germany. Here, we use the 2015 Emissions Database for Global Atmospheric Research (EDGAR v5) (Crippa et al., 2020), which is the most recent year for which data are publicly available. Emissions in EDGAR are provided in Gg/year while pollutant concentrations detected onboard the Zeppelin are measured in ppbv. A direct comparison of the emission inventories and observations is challenging. From emission to detection, measured pollutant concentrations can drastically change due to dilution as well as chemical and physical loss processes whereas the annual inventory emission estimates may differ from daily or even hourly emission rates. A common strategy to reduce the above described uncertainties has been to focus on pollutant mass ratios (Gkatzelis et al., 2021b; Coggon et al., 2021). For example, if CO and a volatile organic compound are co-emitted from a pollution source their ratio is constant as they travel downwind of the source if their physical and chemical loss pathways are not different and they do not have other sources.

Figure 5 shows the diurnal variability of the $NO_x$ to CO slope for all Zeppelin flights between 250 m and 450 m. A sensitivity analysis was performed to derive the observed $NO_x$ to CO slope by applying a linear fit function every 60 s, 100 s, or 1000 s. Linear fits obtained from each time step were further filtered depending on the goodness of fit with data discarded if the coefficient of determination $R^2$ of the linear fit was below 0.6, 0.7, or 0.8, respectively. An overview of this sensitivity analysis to the different time steps and $R^2$ thresholds is given in Figure S8. Overall, the $NO_x$ to CO slopes were within $\pm$ 0.05 g g$^{-1}$ independent of the chosen time steps and $R^2$ thresholds. Therefore, for Figure 5 we choose a 1 min time step and an $R^2$ threshold of 0.6 to discuss the observed variability of the $NO_x$ to CO slopes. Figure 5 also shows the $NO_x$ to CO emission ratios from numerous pollution sources based on the EDGAR 2015. The EDGAR emission inventory was separated into different emission sectors including transportation, industry, building and miscellaneous, and other sources by lumping the IPCC emission categories. Road transportation with no resuspension is the dominant source of $NO_x$ in the inventory accounting for 40 % of the $NO_x$ emitted in Germany with a $NO_x$/CO ratio of 0.68 g/g. Other transportation categories are not expected to contribute more than 6 % to the total $NO_x$ emissions, however they have drastically higher $NO_x$/CO ranging from 3–10 g/g. Industrial emissions are dominated by the categories "main activity electricity and heat production" and "manufacturing industries and construction" accounting for 22 % and 18 % of the total $NO_x$, with $NO_x$/CO at 1.37 and 0.56, respectively. The remaining industrial emissions account for 3 % of the total $NO_x$ and their $NO_x$/CO ranges from 0.017–24.32 g/g. Building and miscellaneous sources account for 11 % of the total

NO$_x$ in EDGAR and are predominantly related to residential emissions and off-road vehicles with NO$_x$/CO at 0.08. NO$_x$ to CO slopes during the Zeppelin flights were relatively constant with a daily average of 0.36 (± 0.03) g/g. These values were in the range of the average emission ratio of road transportation, building and miscellaneous emissions, and specific industrial sources when compared to EDGAR 2015. Higher NO$_x$ to CO levels by a factor of 2 to 5 compared to the daily average were evident promoting sporadic detection of high NO$_x$ emission sources compared to CO. Figure 6 shows that the spatial distribution of these higher NO$_x$ to CO emissions is predominantly related to petroleum refinery and chemical industries along the flight tracks in North Rhine Westphalia and Hessen further promoting the potential of Zeppelin flights to locate different pollution sources in space and time. A characteristic example of higher NO$_x$ to CO industrial emissions is highlighted in section 3.4 and further investigation of individual emission sources is the focus of a future study.

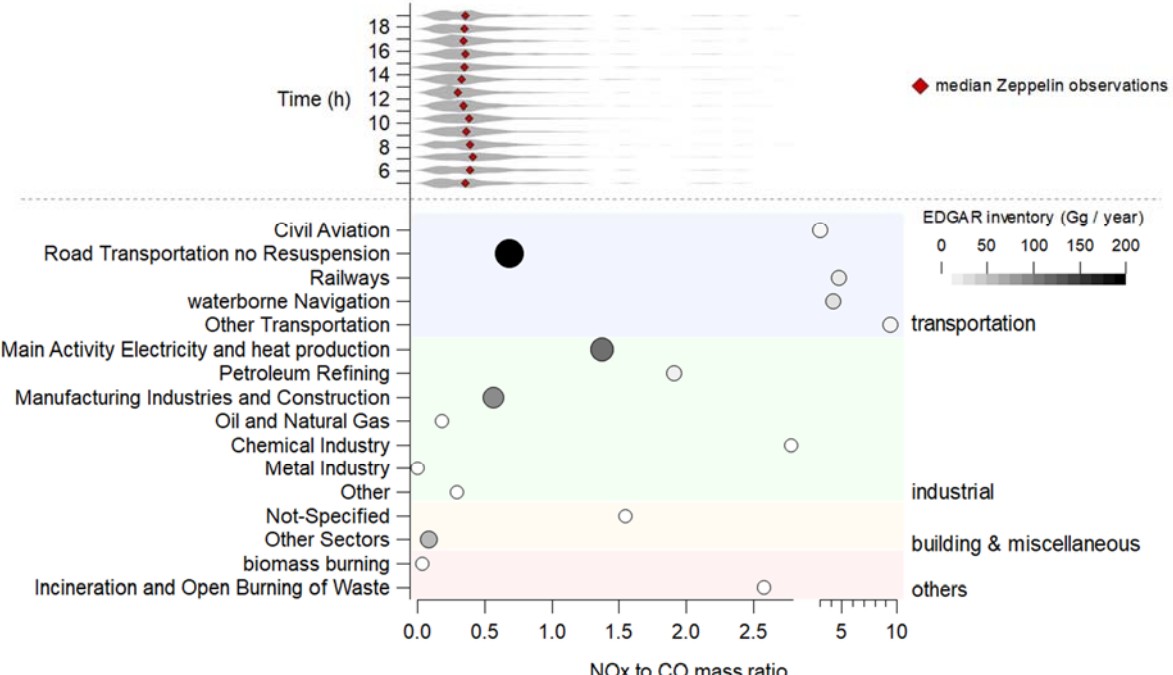

**Figure 5:** Distribution of the NO$_x$ to CO ratio (g / g) during the Zeppelin flights shown as a violin plot for measurements ranging from 250 to 450 meters in height. These ratios are compared to the ratios of different pollution sources following the EDGAR 2015 emission inventory. The size and color of the markers indicate the NO$_x$ emission strength for each EDGAR source.

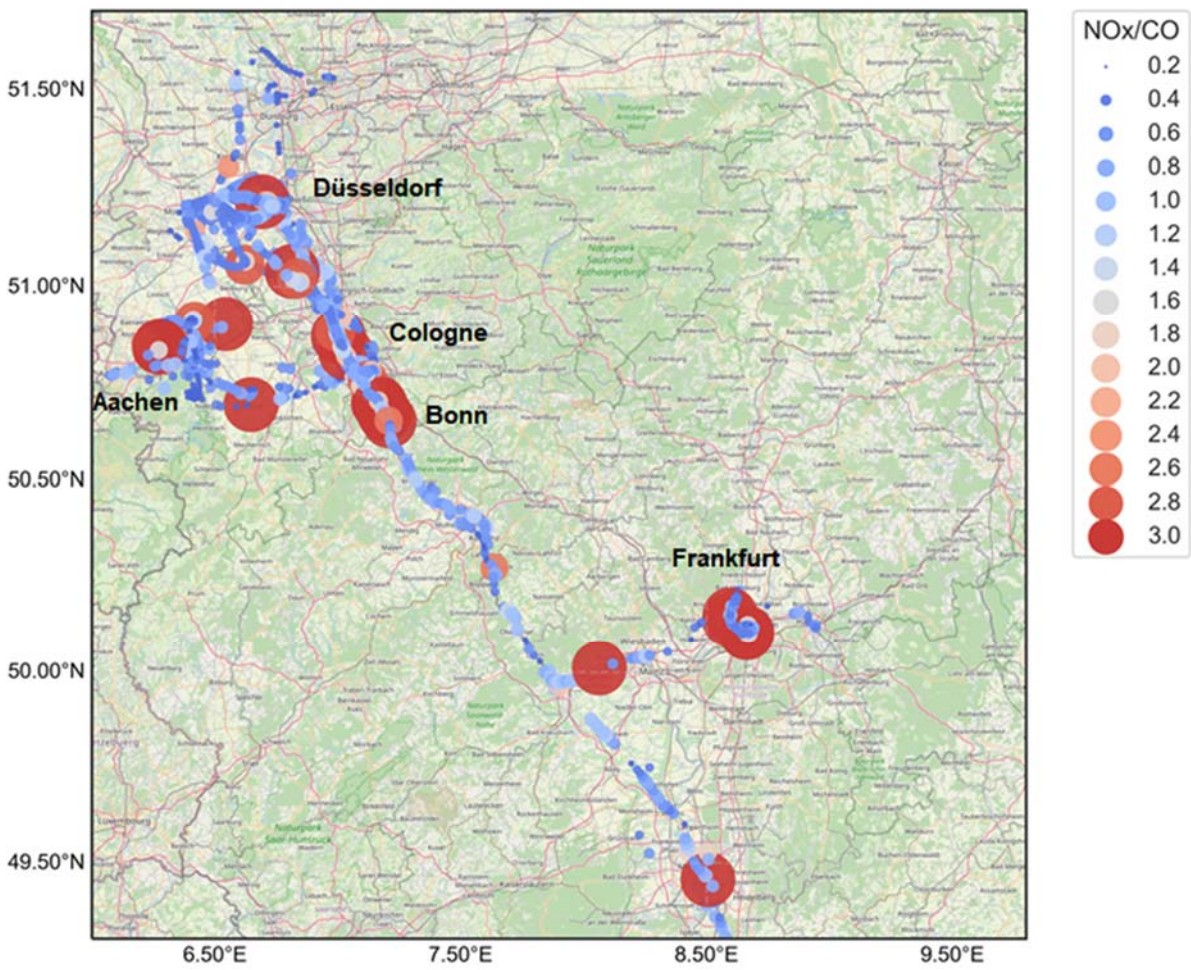

**Figure 6:** Identified areas of high NO$_x$ relative to CO emissions in Germany during the Zeppelin flights (© OpenStreetMap contributors 2021. Distributed under the Open Data Commons Open Database License (ODbL) v1.0.).

The observed NO$_x$/CO slope can be influenced by the longer lifetime of CO compared to NO$_x$ (Seinfeld, 2006) and therefore bias the measurements low. The longer CO lifetime also leads to higher background levels; however, when focusing on the NO$_x$/CO slope this background is accounted for in the offset of the linear fit. The major NO$_x$ daytime loss pathway is the reaction of NO$_2$ with the OH radical yielding nitric acid. The net chemical loss of

NO$_x$ in the atmosphere is challenging to directly observe. Observational methods to determine the lifetime of NO$_x$ have shown that under typical midday conditions in an isoprene-dominated forest it was 11 h ± 5 h (Romer et al., 2016), whereas for studies focused on the outflow of isolated emissions the average range of NO$_x$ lifetimes was around 5-8 h (Ryerson et al., 1998; Liu et al., 2016; Dillon et al., 2002; Alvarado et al., 2010; Valin et al., 2013). The vertical trajectory time for emissions to be detected onboard the Zeppelin is expected to be below the above

lifetime thresholds due to the midday vertical mixing with vertical wind speeds in the range of 1 to 2 m/s (Stull, 1988). However, the net chemical loss of NO$_x$ cannot be neglected, especially if shorter NOx lifetimes are evident due to the efficient production of multifunctional nitrates (e.g., Valin et al., 2013). Therefore, the observational ratios presented here are a lower estimate compared to the ratios close to the emission source.

Uncertainties can also exist in the 2015 EDGAR emission inventory estimates compared to the expected emissions
in 2020 when the Zeppelin flights were performed. The European Environmental Agency reports a drastic
decrease in $NO_x$ emissions by more than 60% from 1990 to 2017 (EEA, 2019). Assuming the decrease in $NO_x$
emissions is stronger compared to the decrease of CO from 2017 to 2020, this could lead to an overestimation of
the EDGAR $NO_x$/CO ratios presented here. Furthermore, 2020 was the year that the COVID-19 pandemic led to
unprecedented government restrictions to limit the spread of the disease. Gkatzelis et al. (2021a) show that reduced
$NO_x$ and CO emissions correlate with stricter government responses that could affect the $NO_x$ and CO
observations. Although transportation and industrial emissions are expected to decline during lockdown
conditions, building emissions could have risen. The increased contribution of building emissions could therefore
explain why observational $NO_x$/CO ratios fall along with the EDGAR transportation/industry and building
emission ratios.

**3.4 Targeted flights to identify coal power plant emissions**

As highlighted in the previous section, emissions from industrial energy production are a major contributor to
pollutant concentrations in Germany. Industrial pollution sources were identified in proximity to the Zeppelin
airports using the EURAD-IM model prior to flights performed in North Rhine-Westphalia. The coal power plant
in Weisweiler (50.838° N, 6.321° E) was chosen to investigate the vertical and horizontal evolution of industrial
emissions on May 8, 2020, at 06:00-14:00 UTC.

Figure 7 shows the EURAD-IM model results (see section 2.4) for $NO_x$ and CO concentrations overlaid with
observations on-board the Zeppelin for one transect crossing the industrial plume at 7:30-8:30 UTC. Comparison
of the EURAD-IM model and observations at different industrial plume heights and different time periods are the
focus of future work; here we highlight the potential of Zeppelin measurements as valuable input for model
evaluation or data assimilation approaches. EURAD-IM $NO_x$ concentrations were at 25 ppbv close to the
industrial emissions and decreased moving downwind of the industry due to dilution. The Zeppelin flew in circles
around the industry during the same period. $NO_x$ background concentrations were at 5-10 ppbv and increased to
10-25 ppbv when flying through the industrial plume. Model concentrations ranged from 5 to 10 ppbv upwind of
the industry location in the southern and eastern regions and were in good agreement with observational trends.
The model $NO_x$ concentrations were on average 18-20 ppbv for this industry and agreed within 12% with the
Zeppelin measurements for this transect. Industrial CO emissions were minimal both in the model and
observations and background CO levels agreed within less than 10%.

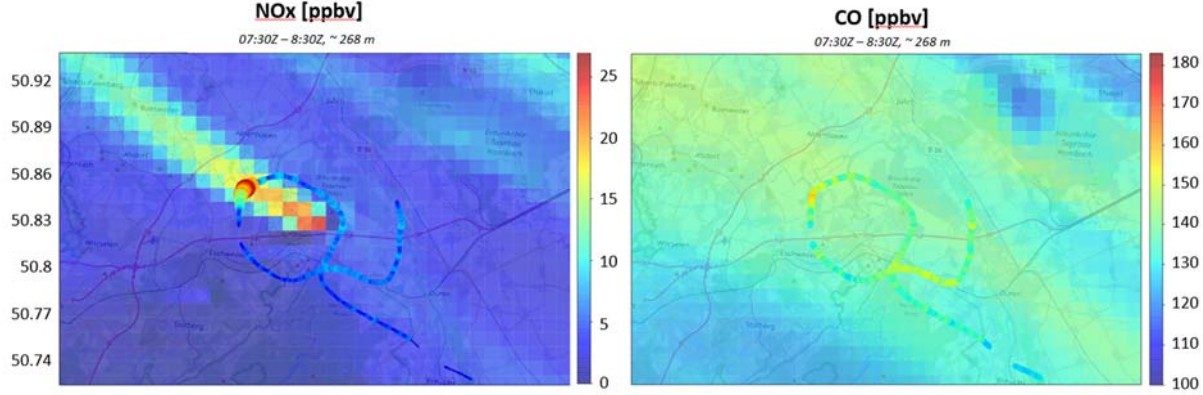

**Figure 7:** Evolution of industrial plume emissions of $NO_x$ and CO based on the EURAD-IM model overlaid by the field-derived Zeppelin observations.

Industrial emissions during the period of the measurements were lower than the business-as-usual emission scenario due to the increased contribution of solar and wind energy (Energy-Charts) and/or the impact of lockdown measures. Particularly, the net electricity generation from the Weisweiler coal power plant during the Zeppelin flight was 50% less compared to the weekly average generation for 2020. The good agreement between the model and observations for $NO_x$ concentrations could therefore be due to an overestimation of $NO_x$ emissions at this given time by the model. However, the model NOx and CO background concentration levels are in good agreement with observations. Although the period of the measurements was during lockdown conditions, the background $NO_x$ and CO levels seem to be unaffected by the stay-at-home orders.

## 4 Conclusions

We report in-situ measurements of air pollutant concentrations within the planetary boundary layer on board the Zeppelin NT airship in Germany. A novel quantum cascade laser-based multi-compound gas analyzer (MIRO Analytical AG) is deployed to simultaneously measure the concentration of greenhouse gases ($CO_2$, $N_2O$, $H_2O$, and $CH_4$) and air pollutants (CO, NO, $NO_2$, $O_3$, $SO_2$, and $NH_3$) with high precision at a measurement rate of 1 Hz. Electrochemical sensors for NO, $NO_2$, $O_x$ ($NO_2+O_3$), and CO, an optical particle counter, temperature, humidity, altitude, and position monitoring are attached to a hatch box below the Zeppelin cabin. In total, 14 commercial flights, 4 targeted flights, and 6 transect flights are performed in May, June, and September 2020, and include flights over urban, remote, but also industrial areas, and highways.

Vertical profiles of pollutants are obtained with a focus on $NO_x$ and $O_3$ during the airship landing and take-off close to the airports of Bonn and Frankfurt. In the early hours from 06:00 to 08:00 UTC, $NO_x$ mixing ratios are higher close to the ground and sharply decrease when above 125 m. $O_3$ has the opposite trend with low mixing ratios close to the ground and an $O_3$ increase above 125 m. This is due to a developing convective mixing layer in which dilution of fresh emissions is less pronounced leading to an increase in $NO_x$ concentrations close to the ground, and the subsequent titration and decrease of ground-level $O_3$. From 08:00 to 10:00 UTC, an increase of $NO_x$ mixing ratios is evident at all heights due to the morning rush hour traffic emissions. From 10:00 to 18:00

UTC, the convective mixing layer is fully developed within the max. flight height of 450 m above ground and well mixed with $NO_x$ and $O_3$ concentrations agreeing within their variability at all heights. During these periods, $NO_x$ concentrations decrease throughout the day due to vertical dilution whereas $O_3$ increases due to increased photochemical activity. We compare the diurnal variability of the Zeppelin vertical profiles to measurements from ground-based monitoring stations in Frankfurt to find that Zeppelin vertical concentrations are predominantly affected by suburban emissions and only the higher altitude measurements are influenced by urban Frankfurt emissions.

$NO_x$ to CO slopes at higher altitudes (250–450 m) are compared to the 2015 EDGAR emission inventory. A daily average of 0.36 (± 0.03) g/g is found for the Zeppelin measurements in the range of the inventory emission ratios for transportation, residential emissions, manufacturing industries, and construction activity. Sporadic high $NO_x$ to CO slopes (2-5 g g$^{-1}$) close to industrial sources are observed including a coal power plant in Weisweiler. We compare dedicated measurements in this industrial facility to the EURAD-IM model to find agreement within less than 15% for $NO_x$ and CO concentrations during one plume transect. However, due to the increased contribution of solar and wind energy and/or the impact of lockdown measures the power plant was operated at 50% capacity; therefore, possible overestimation of emissions by the model cannot be excluded. Nevertheless, an agreement between the model and observations for background $NO_x$ and CO concentrations promotes that emissions were not drastically affected due to lockdown restrictions as they are adequately represented by the model calculations, in which no emission reductions to account for the lockdown have been included.

From obtaining vertical pollutant distributions to evaluating emission inventories and modeling efforts the findings of this work highlight the unique scientific insights obtained aboard a Zeppelin platform and promote the importance of frequent airship measurements in Europe in the following years. Furthermore, the low costs of commercial flights provide an affordable and efficient method to improve our understanding of changes in emissions in space and time. Future efforts to include volatile organic compound measurements along with the greenhouse gases and air pollutants obtained by the MIRO MGA[10]-GP multi-compound gas analyzer will further expand the capabilities of this platform and provide insights into primary and secondary pollution observations.

**Data availability**

Tillmann, Ralf, 2022, "Replication Data for: Zeppelin flights 2020: Air quality observations",
https://doi.org/10.26165/JUELICH-DATA/7ZZIXJ

**Author contribution**

RT, FR and AKS designed the experiments and flight campaigns. BW, CW, TS and FR carried them out. OA and MH provided profound instrumentation support AL, PF and EF provided the model data. GIG, TS and MD visualized the data. GIG and RT prepared the manuscript with contributions from all co-authors. RW, FR, PF and AKS commented on the manuscript.

**Acknowledgments**

We acknowledge the support of Deutsche Zeppelin Reederei (DZR) and Zeppelin Luftschifftechnik GmbH (ZLT); MOSES for funding; Jeff Peischl and Matthew M. Coggon for fruitful discussions and IGOR software development support. The authors gratefully acknowledge the computing time granted through JARA on the supercomputer JURECA at Forschungszentrum Jülich.


The authors declare that they have no conflict of interest.

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
