# Peer review of "Air quality observations onboard commercial and targeted Zeppelin flights in Germany - a platform for high-resolution tracegas and aerosol measurements within the planetary boundary layer"

_Atmospheric Measurement Techniques, 2021_

## Author Comment (AC1)

**Response to Reviewers**

**Air quality observations onboard commercial and targeted Zeppelin flights in Germany - a platform for high-resolution trace-gas and aerosol measurements within the planetary boundary layer**
* * *
Referee comments are in **black** and authors responses are in **blue**. Lines refer to

**Reviewer 1**

Tillmann et al. present observations of greenhouse gases and important primary and secondary pollutants that impact air quality. These measurements were conducted on a Zeppelin, allowing for both targeted measurements as well as measurements conducted at lower altitudes than typical for aircraft. With the observations, the authors were able to show the collapse of the nocturnal residual layer and the mixing of the boundary layer with the residual layer, some investigation into emission sources, and comparisons of emissions/transport with a model for selected sources.

Though the paper is of interest and provides valuable results/information, currently as presented, the paper seems to be more geared for Atmospheric Chemistry and Physics and not Atmospheric Measurement Technique. The reasoning behind that is discussed in more detail below. Further, there are areas where further clarification and discussion about the methodology is needed. Depending on what direction the authors decide to take will make this paper either acceptable for AMT or ACP.

As this was submitted to a measurement technique journal, it would be expected that more details would be included concerning the measurement and techniques. However, the authors provide minimal information about the measurement techniques. The following, at minimum, should be included to make this paper more appropriate for AMT:

We thank the reviewer for the positive feedback and the helpful comments. We understand the reviewer's dilemma on whether this paper should be better geared for ACP given that our study includes both elements of an atmospheric measurement technique by using the Zeppelin as an airborne platform but also the potential of such measurements to understand emissions and their vertical distribution. We think that AMT is the best choice for its publication given that there is no study so far that promotes the capabilities of the Zeppelin for measurements within the planetary boundary layer which we try to highlight as also promoted by our title. Nevertheless, we agree that additional insights on the performance of the commercially available MIRO instrument for all compounds together with a more detailed discussion on the performance of the low-cost sensors would be a valuable addition in this paper. We now include a more detailed discussion on the performance of the instruments and their limitations. Response to each comment is provided below.

1) Comparison of the low-cost sensors along with MIRO. Including, but not limited to, sensitivity, response times, interferences (including any potential RH or temperature dependencies, esp. if this unit is not in a temperature controlled area of the Zeppelin), any pressure dependencies, how limit of detection may change with integration time, etc.

The performance of the low-cost sensors and their optimization are the subject of a different paper we are currently writing and plan to also submit in AMT. We emphasize this at the very end of chapter 2.3 which has been revised accordingly. Also, we now include more information on a comparison of the ECS with the MIRO device in a figure in the supplement (Figure S3) and in the main text as follows:

"Furthermore, the potential of the ECSs to measure nitrogen oxides is shown in Figure S3. On average, ECS NOx data are higher by 20% compared to the MIRO for concentrations above 15 ppbv which is the limit of detection for NOx measured by ECS. This makes the ECSs ideal for the identification of high NOx emission sources during the Zeppelin flights but limited in determining NOx variability at low-NOx environments. Calibrations, sensitivity analysis, and associated uncertainties of the ECSs measurements will be further discussed in a separate publication and are not the focus of this work."

2) Response time of MIRO, along with the lines used. This is especially important as:

      a) MIRO is measuring sticky molecules (H2O and NH3) so how quickly can MIRO actually respond to entering/exiting boundary layer?

      b) As one of the selling points of this package is that it can better characterize boundary layer, residual layer, and potentially free troposphere, the instrument quickly responding entering & exiting these different regimes is important. This can also have potential relative humidity dependencies that will be important to discuss and outline.

We agree with the reviewer that these are important information to include in the main text. We now include a paragraph to describe in more detail the instrument response times based on laboratory measurements for $NH_3$ and $H_2O$ that are also further supported by figures in the supplement (Figure S1). We also discuss current limitations in measuring $SO_2$ due to the high limit of detection. This is now included in section 2.2 as follows:

"The performance of the MIRO with its sampling line to measure sticky molecules including $NH_3$ and $H_2O$ was further examined by laboratory measurements which mimicked the conditions during the Zeppelin flights. Fast changes of pollutant concentrations were applied to determine the response times ($t_{90}$) of the measurement system, which were 240 s and 9 s for $NH_3$ and $H_2O$, respectively (see Figure S1). This highlights the future need for a heated sampling line in order to provide quality assured data, especially for $NH_3$. We therefore omitted $NH_3$ from our discussion. For $H_2O$ and less sticky molecules response times below 9 s result in a spatial horizontal resolution of $< 150$ m considering a horizontal Zeppelin flight speed of 60 km/h and a vertical resolution of $< 15$ m for a vertical speed of 1.7 m/s. This provides the upper limits of the spatial resolution of pollutant concentration but is sufficient for the analysis included in this work. Finally, the instrument detection limit for $SO_2$ is 1.7 ppb i.e., 4.9 µg/m³ (1 σ) and the expected $SO_2$ concentrations in European urban areas are mostly below 5 µg/m³ (Henschel et al., 2013). Therefore, $SO_2$ measurements are omitted from further discussions within this paper."

3) Discussion of the sampling scheme--what is the tubing (type, internal diameter, length) from the inlet to the different instruments? What is the residence time? Is it heated? Similarly for the zero air supply?

This is now added to the main text as follows:

"The sample inlet line connected to the MIRO consisted of an unheated 8 m long PFA (perfluoroalkoxy alkane) tube with an internal diameter of 4 mm. The sample air was drawn from the inlet located at the hatch box below the Zeppelin cabin at a flow rate of 1.2 lpm resulting in a residence time of around 5 s."

and for the zero air supply as follows:

"During the Zeppelin flights, the instrument's inlet was switched to a zero-air supply of $NO_x$- and $O_3$-free air by sampling cabin air through zero air cartridges for 2 min every 20 min."

4) What is the cooling system for? What is the zero air cartridges for? What is the pressure controller for? What happens without these devices?

As mentioned in the main text the cooling system keeps the quantum cascade laser of the MIRO at a constant temperature. Zero air cartridges are used for sequential zeroing of NO, $NO_2$, and $O_3$ during the flights and the pressure controller maintains a constant pressure in the cavity cell of the MIRO. Without these devices we have signal drifts which we now highlight further in the revised main text of chapter 2.2 as follows:

"The Herriott cell is constantly flushed with the sampled gas providing an online in-situ measurement. The pressure in the cell was maintained at 95 hPa using a pressure controller in combination with a membrane pump. For temperature stabilization of the QCLs, an external water chiller is connected to the instrument to minimize drifts caused by ambient temperature variations. During the Zeppelin flights, the instrument's inlet was switched to a zero-air supply of $NO_x$- and $O_3$-free air by sampling cabin air through zero air cartridges for 2 min every 20 min. The obtained zero points were used to apply a background correction to the NO, $NO_2$, and $O_3$ data to reduce their drift."

5) How reliable is this instrument on maintaining the lines/features to measure the compounds listed? How easy is it to get back to the lines/features if the instrument loses them due to pressure or temperature fluctuations?

Pressure and temperature fluctuations are sufficiently removed due to the use of a temperature and pressure controller (see also 4)).

6) An important question that the authors brought up includes boundary layer and residual layer. Currently there are minimal measurements in the residual layer and in differentiating boundary layer, residual layer, and free troposphere. Is there an algorithm from the observations the authors presented that they can estimate the heights of these levels?

We agree with the reviewer that obtaining an algorithm that estimates the heights from the observations would be of great value. We differentiate the boundary layers visually in Fig. 3 by focusing on the maximum increase of Δ(concentration)/Δ(height). An algorithm could be derived following this approach but the limitation here is that our Zeppelin measurements are limited to a few days during the summer period and to specific locations close to the airports of Frankfurt and Bonn. Therefore we consider the visual approach sufficient, given the number of profiles observed.

The following clarifications/discussions would improve the paper:

1) It is currently unclear throughout Section 3 if the authors are only talking about MIRO or if it is a combination of MIRO and electrochemical sensors.

As mentioned above, the focus of section 3 (Results and Discussion) is on the MIRO datasets exclusively. We make that clearer now at the end of chapter 2.3 as follows:

"In the following, all measured pollutant concentrations are acquired from the MIRO instrument."

2) As a side note, since the authors brought up the electrochemical and optical measurements, at least a minor discussion of these measurements would be beneficial for this manuscript. Further, for the optical counter, what type of line (Teflon vs copper/stainless steel) and dryer used?

We emphasize the use of the Zeppelin as a platform for atmospheric measurements. This is valid for both gas- and particle-phase measurements. Currently, the optical counter has been used based on the commercially available recommendations and no further quality assurance has been performed. We therefore excluded these measurements and adapted the main text accordingly including the following sentences:

"Currently, the optical counter has been used based on the available instrument recommendations with and without an isokinetic inlet. The isokinetic inlet has been a 3-D-printed L-shaped inlet of 10 cm length and an internal diameter of 5 mm. The second OPC was used without any inlet line, sampling perpendicular to the flight direction. No further sample preparation or quality assurance has been performed. We therefore exclude the OPC data in its current state from further discussion."

Regarding the electrochemical sensors we now include a brief discussion and a figure given that a more detailed characterization and analysis are the focus of a follow-up publication (chapter 2.3):

"Furthermore, the potential of the ECSs to measure nitrogen oxides is shown in Figure S3. On average, ECS $NO_x$ data are higher by 20% compared to the MIRO for concentrations above 15 ppbv which is the limit of detection for $NO_x$ measured by ECS. This makes the ECSs ideal for the identification of high $NO_x$ emission sources during the Zeppelin flights but limited in determining $NO_x$ variability at low-$NO_x$ environments. Calibrations, sensitivity analysis, and associated uncertainties of the ECSs measurements will be further discussed in a separate publication and are not the focus of this work."

2) As geostationary satellites are coming on-line, a discussion in how this package could be used to validate the geostationary satellites would be of use. This is especially important as the package includes many of the species that the satellites will be trying to target. How does the integrated column change between the different times the authors discuss, along with the air mass factor, between having a nocturnal residual layer and a well mixed boundary layer?

We thank the author for this great point. Comparing satellite observations to ground-based monitoring stations, aircraft measurements, and here the Zeppelin measurements is certainly valuable. The presented Zeppelin measurements cover only the lower PBL up to a height of 800 m. Therefore, we may miss out a significant part of the integrated column. We consider a discussion on the change of the integrated column to be beyond the focus of this AMT paper. Nevertheless, we now include in chapter 3.2 a sentence to further highlight the potential of such comparisons in the future as follows:

"Furthermore, comparison of the Zeppelin flights not only to ground-based monitoring stations but also to modeling efforts as well as satellite measurements would be of great value as geostationary satellites come on-line in the future."

3) The purpose of Fig. 4 and the associated discussion is currently not clear. Numerous studies from NASA DISCOVER-AQ and KORUS-AQ have indicated that NOx is not well mixed between ground and at altitude measurements (e.g., Flynn et al., 2014; Flynn et al., 2016; Choi et al., 2020; Li et al., 2021). A discussion in how the profile from the ground-based observations and the Zeppelin profile may be more suitable.

We include two paragraphs that discusses in more detail the take-home message from Figure 4 and better link it to previous publications including the following:

"For higher altitude measurements (250–300 m) the NOx concentrations peak at 10:00–12:00 UTC which highlights the effect of PBL dynamics where morning emissions are distributed vertically to generate a well-mixed layer, and the dilution of the mixed air masses. This results in a later and weaker peak of the NOx concentration at high altitudes. As the convective layer reaches altitudes above the upper flight range of the Zeppelin this observed change in dynamics is no longer captured. In the afternoon, no significant concentration differences are observed between the different heights. It is striking that the low altitude sub-/urban station measurements and the high altitude, rural monitoring station enclose the Zeppelin observations. Therefore, Zeppelin based observations during passenger flights can be related to nearby ground-based measurements if not exceedingly influenced by local sources as it would be the case in street canyons, i.e., inner urban monitoring sites."

And

"These results provide evidence of effective vertical mixing in particular during the afternoon but limited to the heights captured by the Zeppelin. Future studies to convey whether mixing is as efficient at higher altitudes, that has been previously achieved by aircraft studies (e.g., Flynn et al., 2014; Flynn et al., 2016; Choi et al., 2020; Li et al., 2021), will be of great interest. Furthermore, comparison of the Zeppelin flights not only to ground-based monitoring stations, but also to modeling efforts as well as satellite measurements, would be of great value as geostationary satellites come on-line in the future."

4) Why does it take the NOx concentrations to be higher at 250 - 300 m longer than rest of the boundary layer? Does it make sense with mixing/collapse of the residual layer?

We observe the evolution of the convective mixed layer (see Fig. S5) transferring emissions vertically to higher altitudes where the NOx concentrations were previously lower. We further emphasize this result in the main text and link it to previous publications as discussed above.

5) The authors mention that compounds need to have similar loss rates in order to do ratios to understand emission ratios/sources (Section 3.3). However, NO2 has a very short lifetime (shorter than the 7 - 11 hours the authors noted due to the production of peroxy acyl nitrates, PAN, and alkyl and multifunctional nitrates, e.g., Valin et al., 2013). The authors should show what the ratio of NOx to CO from ground monitoring sites are and compare with the Zeppelin measurements to (a) show confidence in the assumptions and ratios they provide and (b) what new information the Zeppelin provides that ground measurements currently may not provide.

We now include a sentence that highlights that the $NO_2$ lifetime is an upper limit as follows:

"However, the net chemical loss of NOx cannot be neglected, especially if shorter NOx lifetimes are evident due to the efficient production of multifunctional nitrates (e.g., Valine et al., 2013). Therefore, the observational ratios presented here are a lower estimate compared to the ratios close to the emission source."

Following the reviewer's comment, we attempted to generate ratios of NOx/CO for monitoring stations focusing on Frankfurt where we have the statistically highest number of flight measurements. The number of monitoring stations that provide both NOx and CO data are limited to two and provide hourly measurements. When applying a linear slope to the NOx and CO datasets the correlation coefficient, $R^2$, for the overlapping periods is lower than 0.3. Nevertheless, the NOx/CO slope obtained by such analysis is around 0.2-0.3 g/g which is at the lower end of what is presented in Figure 5. Given that ground monitoring stations are influenced by local sources a direct comparison to the Zeppelin flights comes with limitations and uncertainties. Although with the Zeppelin we have the unknown chemical history of the air parcels we still have the unique capability to capture the breath of cities which we consider to be valuable beyond any

ground-based monitoring stations given the limited spatial coverage at the ground and the influence of local sources. We therefore consider it sufficient to include the sentence above that clarifies that these results are a lower estimate of the NOx/CO during our flights.

Minor:

1) Please make sure to be consistent about underscoring the x in NOx.

This is now consistently underscored.

2) Please recheck the grammar and capitilzation throughout the manuscript. E.g., World Health Organization is not capitalized and there are many instances were commas would be appropriate to separate a describer (e.g., line 50, "networks e.g. the European Environment Agency EEA together" should be "networks, e.g., the European Environment Agency, EEA, together"). Also, contractions (e.g., don't) should not be used.

We tracked down the grammar and capitalization misspell throughout the manuscript.

3) It is generally recommended that references that use websites should have similar in-line references as papers in that the website reference is listed at the end in the references section.

The citation of websites has been adapted accordingly.

4) Why is the evening rush hour for NOx at such a late time (~8:00-9:00 PM local time). I would expect the rush hour to be between 4:00 and 7:00 PM local time.

This is what has been observed consistently by several ground-based monitoring stations. The morning agreement between the Zeppelin and ground-based observations gives us more confidence concerning these diurnal trends. Therefore, we consider that the evening peak may reflect differences in work patterns for different continents.

5) Fig. 4: why are some of the flight paths thicker?

This is multiple overlayed flight paths. This is now included in the caption of the figure.

6) Fig. 7: I would recommend the observations have a black outline as they are hard to see with the background model results.

We consider the fact that observations and model are not well separated a good indication that they both agree. We tried changing the graph following the recommendations of the reviewer but unfortunately due to the high time resolution of the Zeppelin measurements the graph is dominated by the black outline. Therefore, we decided to keep the format of the graph as is.

---

## Author Comment (AC2)

**Response to Reviewers**

Air quality observations onboard commercial and targeted Zeppelin flights in Germany - a platform for high-resolution trace-gas and aerosol measurements within the planetary boundary layer

Referee comments are in **black** and authors responses are in **blue**. Lines refer to

**Reviewer 2**

Tillmann et al. provide an overview of the utilization of a new Zeppelin research platform equipped with instrumentation for air quality studies. They provide some examples of the unique sampling strategies that can be provided with such a platform (i.e. details of the vertical structure of the boundary layer) and the use of the platform to evaluate an emissions inventory.

Generally I think that what is presented is well done. Since the paper is submitted to AMT, I was expecting some more details on the evaluation of the data quality from the different instruments onboard. As far as I can tell, the measurements presented are only from the MIRO instrument. I think it would be appropriate to have a section discussing the data quality from the less expensive chemical sensors, and comment on their utility for the future. Was the intention of integrating them along with the MIRO to evaluate them, or was it to possibly rely on those only in the future for this or other platforms?

On the MIRO side, not all of the chemical measurements were discussed. I think it would be good to comment on the SO2 and NH3 data quality, since these are important for air quality studies and the use of a single instrument capable of providing all of those measurement would be really of wide interest. As the manuscript is though, we don't know if those measurements were deemed to be of sufficient quality for air quality research.

I think that it may be appropriate for a revised version of the paper to be published in AMT, but think that first the sections that deal with the instrumentation should be expanded a bit to:

Provide some comments on the observed data quality or issues from the MIRO, for example how much did the zeros drift?

Expand the data discussion to comment at least on the SO2 and NH3 measurement and compare the measurements between MIRO and the sensors.

We thank the reviewer for the helpful comments and positive feedback. We agree that more information on the electrochemical sensors and MIRO would add value to this paper. As mentioned to reviewer 1 the performance of the low-cost sensors and their optimization are the subject of a different paper we are currently writing and plan to also submit in AMT. Nevertheless, we now include more information on this in the main text which includes the following:

"Furthermore, the potential of the ECSs to measure nitrogen oxides is shown in Figure S3. On average, ECS NOx data are higher by 20% compared to the MIRO for concentrations above 15 ppbv which is the

limit of detection for NOx measured by ECS. This makes the ECSs ideal for the identification of high NOx emission sources during the Zeppelin flights but limited in determining NOx variability at low-NOx environments. Calibrations, sensitivity analysis, and associated uncertainties of the ECSs measurements will be further discussed in a separate publication and are not the focus of this work. In the following, all measured pollutant concentrations are acquired from the MIRO instrument."

Comments on the observed data quality and issues from the MIRO are dealt with in a previous paper, which has now been included as a reference in chapter 2.2:

"We deployed a MIRO MGA10-GP multi-compound gas analyzer, a newly available commercial instrument (MIRO Analytical AG, Wallisellen, Switzerland). The analyzer measures ten trace gases NO, NO2, O3, SO2, CO, CO2, CH4, H2O, NH3, N2O with a time resolution of 1s and precisions (1 $\sigma$ ) as summarized in Table 2. The stated precisions were determined by Allan-Werle-Variance (Werle et al., 1993). A detailed description of the measurement principle and the instrument's data processing can be found elsewhere (Liu et al., 2018, Hundt et al., 2018)."

Regarding the performance of MIRO in measuring SO2, NH3, and H2O we now include a more detailed discussion on the instrument capabilities in the main text. Given that the MIRO inlet was not optimized for the detection of these compounds these are not further used throughout the manuscript but we rather highlight the potential of a revised setup to measure these compounds in the future.

"The sample inlet line connected to the MIRO consisted of an unheated 8 m long PFA (perfluoroalkoxy alkane) tube with an internal diameter of 4 mm. The sample air was drawn from the inlet located at the hatch box below the Zeppelin cabin at a flow rate of 1.2 lpm resulting in a residence time of around 5 s.

The performance of the MIRO with its sampling line to measure sticky molecules including NH3 and H2O was further examined by laboratory measurements which mimicked the conditions during the Zeppelin flights. Fast changes of pollutant concentrations were applied to determine the response times (t90) of the measurement system, which were 240 s and 9 s for NH3 and H2O, respectively (see Figure S1). This highlights the future need for a heated sampling line in order to provide quality assured data, especially for NH3. We therefore omit NH3 from further discussion. For H2O and less sticky molecules response times below 9 s result in a spatial horizontal resolution of < 150 m considering a horizontal Zeppelin flight speed of 60 km/h and a vertical resolution of < 15 m for a vertical speed of 1.7 m/s. This provides the upper limits of the spatial resolution of pollutant concentration but is sufficient for the analysis included in this work. Finally, the instrument detection limit for SO2 is 1.7 ppb i.e., 4.9 µg/m3 (1  $\sigma$ ) and the expected SO2 concentrations in European urban areas are mostly below 5 µg/m3 (Henschel et al., 2013). Therefore, SO2 measurements are omitted from further discussions within this paper."